# Serum Levels of IL-21 and IL-22 in Breast Cancer Patients—A Preliminary Study

**DOI:** 10.3390/cimb47070537

**Published:** 2025-07-10

**Authors:** Jacek Kabut, Aleksandra Mielczarek-Palacz, Joanna Magdalena Gola, Elżbieta Chełmecka, Anita Gorzelak-Magiera, Patrycja Królewska-Daszczyńska, Sebastian Stępień, Jakub Szymon Wnuk, Iwona Gisterek-Grocholska

**Affiliations:** 1Department of Oncology and Radiotherapy, Medical University of Silesia, 40-055 Katowice, Poland; anitagor@op.pl (A.G.-M.);; 2Department of Immunology and Serology, Faculty of Pharmaceutical Sciences in Sosnowiec, Medical University of Silesia, 40-055 Katowice, Poland; apalacz@sum.edu.pl (A.M.-P.); pdaszczynska@sum.edu.pl (P.K.-D.); sebastian.stepien@sum.edu.pl (S.S.); 3Department of Molecular Biology, Faculty of Pharmaceutical Sciences in Sosnowiec, Medical University of Silesia, 40-055 Katowice, Poland; jgola@sum.edu.pl; 4Department of Medical Statistics, Faculty of Pharmaceutical Sciences in Sosnowiec, Medical University of Silesia, 40-055 Katowice, Poland; ebchelmecka@gmail.com

**Keywords:** breast cancer, IL-21, IL-22, tumour microenvironment

## Abstract

Breast cancer is one of the most commonly diagnosed malignant tumours in women worldwide. Although modern medicine has led to advanced diagnostic methods and therapies that allow for increasingly effective treatment, the mechanisms underlying breast cancer development and progression remain the subject of intensive research. In the pathogenesis of this cancer, significant importance is attributed to interactions between tumour cells and the tumour microenvironment, in which soluble immune system mediators—cytokines—play a key role, including IL-21 and IL-22. These interleukins, by modulating the immune response, can both promote and inhibit tumour progression, and analysing their concentrations may prove helpful in diagnosis, disease progression prognosis, and the development of new therapies, including immunotherapy. The aim of this study was to determine the concentrations of IL-21 and IL-22 in a group of patients with invasive cancer, depending on the biological type of the tumour and its malignancy grade. The study involved 60 women with breast cancer and 20 women with benign breast lesions, and the analysis of IL-21 and IL-22 protein concentrations was performed using the enzyme-linked immunosorbent assay (ELISA) method. The analysis shows that the concentrations of IL-21 and IL-22 do not differ significantly depending on the malignancy grade of the tumour. However, a statistically significant negative correlation between the concentrations of IL-21 and IL-22 was observed exclusively in the group of patients with benign breast lesions. Due to the high heterogeneity of breast cancers, further research with a larger study group is necessary to better understand these parameters and possibly apply them clinically in patients with breast cancer.

## 1. Introduction

Due to its prevalence, breast cancer is a major health and social challenge. Despite significant progress in diagnosis and treatment, it remains one of the leading causes of cancer-related deaths among women [1]. The most important prognostic factors for early breast cancer include the tumour size, the presence and number of metastases in lymph nodes, the histological type and malignancy grade of the tumour, the invasion of peritumoural lymphatic and venous vessels, the Ki67 proliferation index, as well as the status of HER2, ER, and PR receptors. Assessing the malignancy grade of breast cancer helps predict the aggressiveness of the tumour and its tendency to spread. Tumours with a low malignancy grade (G1) have a better prognosis, while highly malignant tumours (G3) are more aggressive and have a worse prognosis.

Contemporary breast cancer research has led to the identification of several major molecular subtypes, which differ not only in their genetic profile but also in their response to treatment and prognosis. The molecular classification of breast cancer is primarily based on the expression of hormonal receptors, HER2 protein status, and the degree of expression of the Ki67 cell proliferation index, and allows for more precisely tailored therapy to fit the individual characteristics of a tumour [2].

The molecular classification of breast cancer not only provides a better understanding of the biology of this cancer but also allows for the personalisation of therapy, which increases the effectiveness of treatment. The division into specific molecular subtypes enables the use of the most appropriate therapeutic methods, such as hormonal therapy, HER2-targeted drugs, chemotherapy, and, in some subtypes, even immunotherapy.

The early diagnosis and identification of prognostic and therapeutic markers are crucial to improving prognosis and tailoring therapy to the individual needs of patients [3]. Despite advances in diagnosis and treatment, the exact mechanisms underlying breast cancer development and progression are not fully understood, as they result from complex interactions between tumour cells and the tumour microenvironment [4,5], anti-tumour response, and the mechanisms that promote tumour development [6,7].

IL-21 is mainly produced by T helper lymphocytes (Th17 and Tfh) and NK cells, whose main role is to modulate the effector functions of the immune system, such as the proliferation, differentiation, and cytotoxicity of T lymphocytes and NK cells [8,9]. IL-21 may both exert anti-tumour activity, by supporting the activation of the immune response, and promote tumour progression, by influencing the tumour microenvironment, including angiogenesis and tumour cell stimulation [8,10].

IL-22, secreted by Th22 lymphocytes, NK cells, and other lymphocyte subpopulations, is a cytokine whose primary role is to modulate the inflammatory response. IL-22 can promote both the growth and invasiveness of cancer cells and exert anti-tumour effects by limiting the chronic inflammation that promotes carcinogenesis [11,12,13]. Studying the levels of IL-21 and IL-22 in the serum of breast cancer patients may contribute to a better understanding of the mechanisms underlying the pathogenesis of this tumour type and offer new perspectives on the diagnosis and therapy of breast cancer, including the identification of biomarkers to support personalised treatment.

Various molecular subtypes of breast cancer are characterised by distinct patterns of immune involvement, including differences in the composition of tumour-infiltrating lymphocytes, cytokine expression profiles, and sensitivity to immunomodulatory treatments. Based on this, we postulated that the serum concentrations of IL-21 and IL-22 might vary depending on the tumour’s molecular subtype and histological grade. Exploring these differences could provide insight into subtype-specific immune interactions and support the identification of novel prognostic or therapeutic immunological markers.

## 2. Materials and Methods

### 2.1. Materials

Patients in both the malignant and benign lesion groups were diagnosed based on the presence of a solid breast tumour detected via imaging tests (breast ultrasound, mammography). At the first visit, medical history was obtained regarding major complaints, chronic diseases (including neoplastic and autoimmune diseases), medications used, and obstetric history. The inclusion criterion for both groups was the presence of a solid breast lesion requiring histopathological verification. Patients with autoimmune diseases or previous oncological treatment were excluded from the study (exclusion criteria).

For all patients, laboratory tests, an ultrasound-guided core needle biopsy of the breast tumour, and a fine-needle biopsy of the axillary lymph nodes (if metastases were suspected) were performed. All benign lesions were histopathologically confirmed. All patients in the malignant group underwent imaging studies to assess for the presence of distant metastases.

The malignant group consisted of 60 patients aged 39 to 87 years (mean ± standard deviation: 64.9 ± 12.8 years) with histopathologically confirmed invasive breast cancer. The histopathological examination included information on the histological type, malignancy grade (G1, G2, G3), receptor status (ER, PR, HER2 expression), and expression of the proliferation index Ki67. Based on the clinical data, the stage of neoplastic disease was assessed according to the TNM classification. Molecular features included in the histopathological protocol allowed the patients to be classified into one of the following breast cancer types: luminal A (20 patients), luminal B HER2-negative (22 patients), luminal B HER2-positive (6 patients), non-luminal HER2-positive (5 patients), and TNBC (7 patients).

The benign group consisted of 20 patients aged 29 to 72 years (mean ± standard deviation: 50.9 ± 13.9 years). Seven patients had fibroadenoma, nine had fibrosclerosis, two had apocrine metaplasia, one had intraductal papilloma, and one had a simple cyst. Although no formal age-matching was performed, the demographic characteristics of both groups, including age distribution, were assessed and are presented to ensure comparability.

The test material consisted of venous blood collected in a tube without anticoagulant (“clot”) and in a tube containing K_2_EDTA (potassium salt of ethylenediaminetetraacetic acid). The collected tube without anticoagulant was set aside for 30 min and then centrifuged at 1500× *g* for 15 min. The serum obtained was aliquoted and frozen at −80 °C. Laboratory, imaging, and histopathological examinations were performed at the Diagnostic Centre of the District Hospital in Rybnik.

### 2.2. ELISA Tests

The IL-21 concentration was determined via immunoenzymatic sandwich ELISA tests using the Human Interleukin-21 ELISA kit produced by BioVendor (Brno, Czech Republic). The sensitivity of this test was 20.0 pg/mL, while the detection range was 78–5000 pg/mL. In this test, the intra-assay precision was 5.8%, while the inter-assay precision was 7.7%.

The IL-22 concentration was determined via immunoenzymatic sandwich ELISA tests using a Human Interleukin-22 ELISA kit produced by BioVendor (Brno, Czech Republic). The sensitivity of this test was 5.0 pg/mL, while the detection range was 31.3–2000 pg/mL. In this test, the intra-assay precision was 6.7%, while the inter-assay precision was 4.5%.

The characteristics of each test used in this study are shown in Table 1.

The principle of the tests is based on the immunological reaction of the binding antigen to the antibody. The tested material was added to a microwell coated with specific antibodies. An antigen–antibody complex was formed through the binding of the antigen to the specific antibody. During washing, unbound biological components were removed. Then, the biotin-conjugated antibody was added and bound to the antigen captured by the first antibody. During the next wash step, the unbound biotin-conjugated antibody was removed. Subsequently, streptavidin-HRP was added and bound to the biotin-conjugated antibody. Similarly, unbound streptavidin-HRP was removed during the wash. The next step involved adding the substrate solution reacted with HRP to the microwells. This process resulted in a coloured product, the intensity of which was proportional to the antigen amount. Adding a STOP solution caused the termination of the reaction. The intensity of the colour in the respective microwells was measured spectrophotographically at a wavelength of 450 nm.

The above-mentioned tests were performed according to the protocol provided by the manufacturer.

### 2.3. Statistical Analysis

An assessment of the normality of the distributions of IL-21 and IL-22 concentrations was performed using a Shapiro–Wilk test and quantile–quantile plots. Due to the lack of normality of the distributions, the median of the lower and upper quartiles was used to describe the data: Me (Q1–Q3). The results are presented in box plots, where the markers correspond to the medians, the boxes indicate the interquartile range, and the “whiskers” extend to the furthest data point, which is within 1.5 times the interquartile range (IQR). Data points further than that distance are considered outliers and are marked with a dot (_o_), while extreme values differ by 3 times the IQR and are marked with an asterisk (*).

We used a non-parametric Mann–Whitney U test when comparing the two groups, and, for a larger number of comparison groups, we used a Kruskal–Wallis ANOVA. In order to assess the relationship between the variables, a Spearman’s rank correlation test was used.

The age variable, presenting a normal distribution, is presented as mean ± standard deviation (m ± s), and a Student’s *t*-test was used to compare the ages of the benign and malignant groups.

A MedCalc calculator was used to compare the coefficients of variation. (MedCalc Software Ltd. (Ostend, Belgium) Comparison of Coefficients of Variation calculator: https://www.medcalc.org/calc/comparison_of_coefficientsofvariation.php (Version 23.1.7; accessed on 1 March 2025)).

Two-sided tests were performed, and the results were considered statistically significant with a significance level of *p* < 0.05. Calculations were performed using Statistica software version 13 (TIBCO Software Inc., Palo Alto, CA, USA (2017)).

## 3. Results

The patients were divided into the following groups based on the degree of malignancy: 12 women (20%) in group G1, 30 women (50%) in group G2, and 18 women (30%) in group G3. Additionally, 20 women constituted the group of patients with benign breast neoplasms.

According to the biological type of cancer, the patients were categorised into five groups: luminal A (20 women, 33%); luminal B, HER2-positive (6 women, 10%); luminal B, HER2-negative (22 women, 37%); triple-negative breast cancer (TNBC) (7 women, 12%); and non-luminal, HER2-positive (5 women, 8%). The details of the study are presented in Figure 1.

The IL-21 and IL-22 concentrations were measured for all subjects; detailed descriptive statistics are presented in Table 2 and Table 3.

The coefficients of variation were compared, and no differences were found for IL-21 (F = 0.599, *p* = 0.145) and IL-22 (F = 0.722, *p* = 0.341).

### 3.1. IL-21 Concentration

There were no significant differences in IL-21 serum levels between the malignant and benign lesion groups (*p* = 0.969).

Similarly, no statistical difference was observed in the IL-21 serum levels between patients with benign breast lesions and those diagnosed with invasive cancer, considering both the grade of tumour malignancy (*p* = 0.573) and the biological type of the tumour (*p* = 0.759) (Figure 2).

### 3.2. IL-22 Concentration

There were no significant differences in the IL-22 serum levels between the malignant and benign lesion groups (*p* = 0.868). Similarly to IL-21, no statistical differences in the IL-22 serum levels were observed when considering the grade of malignancy (*p* = 0.931) and the biological type of the tumour (*p* = 0.601) (Figure 3).

### 3.3. IL-21/IL-22 Concentration Ratio

No differences were found in the IL-21/IL-22 concentration ratio between the group of patients diagnosed with breast cancer and the patients with benign breast lesions (*p* = 0.668). There were also no differences between the groups according to the grade of malignancy (*p* = 0.791) or the biological type of the tumour (*p* = 0.467).

### 3.4. Correlations Between IL-21 and IL-22 Serum Levels

A moderate negative correlation was observed between IL-21 and IL-22 serum levels in patients with benign breast lesions (rS = −0.663, *p* = 0.002), whereas no such correlation was found in patients diagnosed with breast cancer (rS = −0.109, *p* = 0.410).

## 4. Discussion

In recent years, the personalisation of cancer therapies has become a key element in improving the treatment effectiveness, including breast cancer [14,15].

Breast cancer is a heterogeneous disease with distinct molecular subtypes, each characterised by different tumour biology and immune interactions involving immune system mediators—interleukins. These have both pro- and anti-tumour effects. These effects are particularly important in aggressive subtypes, such as triple-negative breast cancer (TNBC) [16,17].

IL-21 is a pleiotropic cytokine that supports anti-tumour immunity but may also facilitate tumour progression [18,19].

Previous studies have shown an overexpression of the IL-21 gene in breast cancer tissue in patients with advanced or metastatic disease [20]. Similar studies were conducted by Xinbin He et al. [21], who assessed both the IL-21 gene expression (qRT-PCR) and the protein levels in the tumour tissue (Western blot) and in a control group. Their results indicate the role of the interleukin in the tumour progression [21]. Other researchers analysed the expression of IL-21 and IL-21R in peripheral immune cells. Their studies did not show a correlation with the stage of the disease [22].

The IL-21 serum concentrations in patients with invasive breast cancer have not yet been analysed, taking into account both molecular subtypes and degree of malignancy.

We conducted such studies for the first time in patients with invasive breast cancer and in a control group consisting of patients with non-malignant tumours. The results showed changes in the IL-21 levels in patients with invasive breast cancer. Although these differences were not statistically significant, the changes observed may suggest the involvement of the interleukin in systemic interactions between immune system cells, both in breast cancer and in benign tumours.

The analysis did not reveal any significant differences in the serum IL-21 concentrations in breast cancer patients, regardless of histological grade or molecular subtype. However, further detailed studies involving larger groups of patients are needed to fully understand these processes.

IL-21 has also been studied in other malignant tumours. Preclinical and early-phase studies have shown promising results for recombinant IL-21 in haematological malignancies and solid tumours [23,24,25,26], confirming its potential as an immunotherapeutic agent.

IL-22 is a cytokine from the IL-10 family produced by Th17 lymphocytes, γδ T lymphocytes, NKT cells, and innate lymphoid cells (ILCs) [27]. A growing number of studies indicate both the protective and pathogenic properties of IL-22 in various diseases, including cancer initiation and progression [28,29].

Xiao et al. demonstrated that IL-22 is overexpressed in TNBC and contributes to tumour proliferation and migration [30]. Zhang et al. [31], using a 4T1 mouse model of breast cancer, showed that IL-22 promotes tumour cell proliferation through a STAT3-dependent mechanism [31]. Similarly, Rui et al. [32] confirmed the involvement of IL-22 and its receptor IL-22R1 in breast cancer progression, reporting elevated serum IL-22 levels and an increased IHC expression of IL-22 and IL-22R1 in tumour tissues compared to normal tissues [32].

In our studies, we also attempted to determine the expression of these interleukins at the mRNA level in whole blood, but we did not obtain confirmation of the presence of these transcripts. This is consistent with data in the database, which did not show the presence of transcripts for IL-21 and IL-22 in normal immune cells circulating in the blood or was limited to specific T-cell subpopulations. In our study, the absence of interleukin mRNA in the blood suggests that breast cancer does not stimulate the expression of these interleukins in circulating immune cells. Therefore, the presence of IL-21 and IL-22 proteins in serum may reflect changes in the tumour microenvironment.

Our study also analysed IL-22 serum concentrations in patients with invasive breast cancer, taking into account the molecular subtype and degree of tumour malignancy, as well as in patients with non-malignant breast cancer.

Similar to IL-21, our results showed changes in the IL-22 levels in patients with invasive breast cancer. Although these differences were also not statistically significant, the changes observed may suggest the involvement of the interleukin under study in systemic interactions between immune system cells, both in breast cancer and in benign tumours. No significant differences in the serum IL-22 concentrations were found in breast cancer patients, regardless of the histological grade or molecular subtype. Further detailed studies involving larger groups of patients are needed to understand the processes involving the interleukin under examination.

An important discovery of our research is the statistically significant negative correlation between IL-21 and IL-22 concentrations in the blood serum of women with benign breast tumours. As the IL-21 concentrations increased, the IL-22 concentrations decreased. This is an interesting and important observation, which indicates that the negative correlation between IL-21 and IL-22 observed in the group of patients with benign tumours may suggest the existence of an intact regulatory mechanism in non-tumour tissue, which is lost with the development of malignant tumours.

In particular, IL-21 and IL-22, although often co-regulated in inflammatory processes, may exert opposite effects depending on the cellular context. In benign lesions, elevated IL-21 levels may promote immune surveillance by enhancing the response of cytotoxic T lymphocytes and NK cells, while a simultaneous decrease in IL-22 may reflect a downregulation of proliferative and tissue repair pathways that are unnecessary in benign conditions. This inverse relationship may also suggest a regulatory feedback loop in which increased IL-21 inhibits IL-22 production or vice versa to maintain immune homeostasis and prevent excessive tissue remodelling or inflammation.

Further studies with larger sample sizes and stratification by lesion type are warranted to confirm and further elucidate these mechanisms. This relationship suggests that significant changes in the immune system, including the dysregulation of cytokine production, occur in the tumour microenvironment. The tumour may affect immune cells and modulate IL-21 and IL-22 secretion independently of natural regulatory mechanisms. Furthermore, the presence of tumour cells and the associated inflammation may lead to atypical cytokine secretion patterns, which may explain the lack of correlation between IL-21 and IL-22 in this group, but further detailed studies are needed.

Unfortunately, our study has several limitations.

The key limitation of this study is the relatively small sample size, especially within individual molecular subgroups of breast cancer. The unbalanced distribution of cases across tumour subtypes reduces the statistical power of subgroup analyses and may limit the generalisability of the results. Therefore, comparisons based on molecular classification should be interpreted with caution.

Furthermore, due to the limited number of patients included in the study and the preliminary nature of the study, the results require validation in a larger cohort. We plan to conduct multicentre studies in the future to improve the statistical reliability and ensure a more representative patient group for the evaluation of IL-21 and IL-22 as potential immune markers in breast cancer.

Another limitation is the inability to perform tissue-level analysis, which prevents deeper analysis. We plan to conduct such studies as part of further analysis in the future, as the results obtained so far provide new and valuable insights into the role of the interleukins studied in the pathogenesis of breast cancer. Our research is composed of preliminary observations that require further and more detailed analysis.

## 5. Conclusions

This preliminary study did not reveal statistically significant differences in serum IL-21 and IL-22 levels between patients with invasive breast cancer and those with benign breast lesions. No associations were found between cytokine levels and tumour grade or molecular subtype. A statistically significant negative correlation between IL-21 and IL-22 was observed only in the group with benign lesions.The small sample size and heterogeneity of the study population likely limited the statistical power of the analysis. Therefore, these findings should be interpreted with caution. Future research involving larger, well-balanced cohorts and multicentre collaboration is necessary to validate the potential role of IL-21 and IL-22 as biomarkers and to better understand their involvement in the breast cancer pathogenesis.

## Figures and Tables

**Figure 1 cimb-47-00537-f001:**
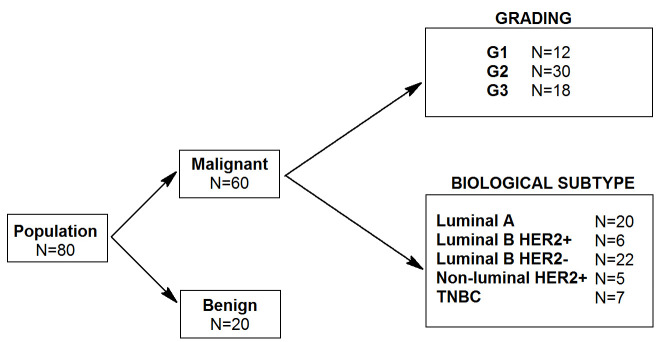
Schematic diagram of the study groups.

**Figure 2 cimb-47-00537-f002:**
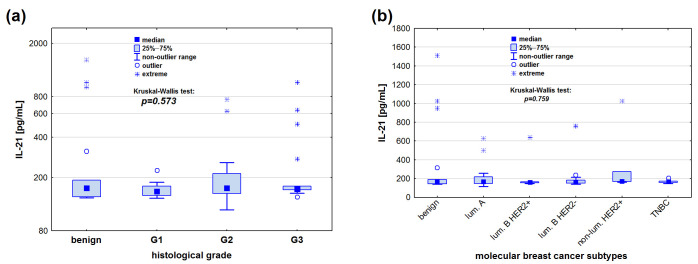
IL-21 serum concentrations [pg/mL] in relation to (**a**) tumour grade and (**b**) biological subtype. Kruskal–Wallis test was used for comparisons. Logarithmic scale applied in panel (**a**) for better visualisation.

**Figure 3 cimb-47-00537-f003:**
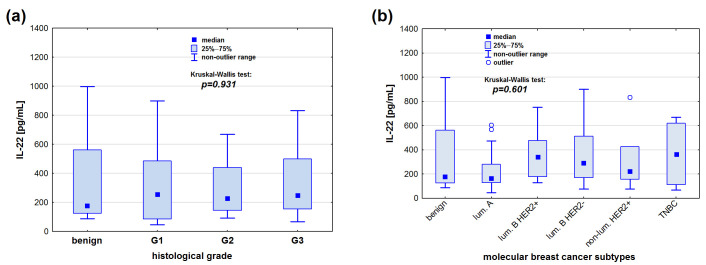
IL-22 serum concentrations [pg/mL] in relation to (**a**) tumour grade and (**b**) biological subtype. Kruskal–Wallis test was used for statistical analysis.

**Table 1 cimb-47-00537-t001:** Laboratory specifications of the tests used in the study. The information below was obtained based on the protocols provided by the manufacturer.

Parameters	Test Name	Unit	Sensitivity	Detection Range	Precision	
					Intra-Assay	Inter-Assay
IL-21	Human Interleukin-21 ELISA manufactured by BioVendor	pg/mL	20.0	78–5000	5.8%	7.7%
IL-22	Human Interleukin-22 ELISA manufactured by BioVendor	pg/mL	5.0	31.3–2000	6.7%	4.5%

**Table 2 cimb-47-00537-t002:** Descriptive statistics of Il-21 [pg/mL] levels in the malignant group with consideration of the group’s malignancy grade and biological type.

Variables	Group	N	m	s	Me	Q_1_	Q_3_	min	max	Vs	As
**IL-21 [pg/mL]**	All groups/ General	78	244.1	239.6	164.8	152.0	190.5	114.8	1509.1	98.2	3.46
	Benign	19	325.7	387.7	166.9	143.8	190.5	141.1	1509.1	119.0	2.31
	Malignant	59	217.9	163.5	164.1	153.3	205.4	114.8	1026.7	75.0	3.47
	**Malignant group based on degree of malignancy**										
	G1	12	164.0	23.6	158.4	147.3	172.3	140.5	226.4	14.39	1.84
	G2	29	212.9	139.8	166.9	152.0	214.9	114.8	763.6	66.66	3.30
	G3	18	261.9	233.5	163.5	161.4	172.3	143.2	1026.7	98.16	2.61
	**Malignant group based on biological type**										
	Lum. A	20	211.2	128.3	165.5	146.2	219.3	114.8	629.6	60.74	2.63
	Lum. B HER2+	6	238.0	196.9	159.4	156.0	164.1	149.3	639.8	82.73	2.45
	Lum. B HER2−	21	200.7	132.3	162.1	152.6	184.4	140.5	763.6	65.93	4.22
	Non-lum. HER2+	5	361.0	375.2	172.3	167.5	276.5	162.1	1026.7	103.9	2.15
	TNBC	7	169.0	18.2	166.9	158.0	171.6	147.2	205.4	10.76	1.42

Legend: N, number of cases; m, mean; s, standard deviation; Me, median; Q_1_, lower quartile; Q_3_, upper quartile; min, minimum value; max, maximum value; Vs, coefficient of variation; As, skewness.

**Table 3 cimb-47-00537-t003:** Descriptive statistics of Il-22 [pg/mL] levels in the malignant group with consideration of the group’s malignancy grade and biological type.

Variables	Group	N	m	s	Me	Q_1_	Q_3_	min	max	Vs	As
**IL-22 [pg/mL]**	All groups/ General	80	316.8	231.3	242.4	131.5	473.9	45.9	997.9	73.0	1.03
	Benign	20	336.9	291.5	178.1	124.5	561.2	87.9	997.9	86.5	1.08
	Malignant	60	310.1	209.9	244.2	140.3	466.0	45.9	900.1	67.7	0.90
	**Malignant group based on degree of malignancy**										
	G1	12	318.9	269.2	255.5	87.0	485.2	45.9	900.1	84.42	1.05
	G2	30	288.6	176.4	226.7	145.4	439.0	93.1	669.5	61.11	0.75
	G3	18	340.2	226.2	248.6	156.0	498.3	68.3	833.7	66.49	0.78
	**Malignant group based on biological type**										
	Lum. A	20	229.7	157.5	166.5	129.8	279.2	45.9	606.6	68.56	1.39
	Lum. B HER2+	6	370.3	233.0	343.8	178.7	475.6	128.0	751.6	62.93	0.80
	Lum. B HER2−	22	340.4	212.3	290.5	167.0	510.6	77.4	900.1	62.37	0.81
	Non-lum. HER2+	5	342.3	303.9	218.9	156.0	426.7	76.1	833.7	88.79	1.37
	TNBC	7	370.3	237.8	363.8	110.6	620.6	68.3	669.5	64.22	−0.03

Legend: N, number of cases; m, mean; s, standard deviation; Me, median; Q_1_, lower quartile; Q_3_, upper quartile; min, minimum value; max, maximum value; Vs, coefficient of variation; As, skewness.

## Data Availability

The data presented in this study are available on request from the corresponding author.

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
