# Peer review of "Serum Levels of IL-21 and IL-22 in Breast Cancer Patients—A Preliminary Study"

_cimb, 2025, doi:10.3390/cimb47070537_

Round 1
Reviewer 1 Report
Comments and Suggestions for Authors
Dear authors,
Thank you for your submission. Your manuscript addresses a relevant topic regarding immune system mediators in breast cancer. However, several methodological and interpretational aspects limit its impact and clarity. Below are specific suggestions and concerns to consider:
-Although the cohort is described as 80 patients (60 malignant, 20 benign), Table 2 includes only 78 (59 + 19). Please explain why two samples were excluded from the IL-21 analysis.
-The analysis stratifies patients into six molecular subtypes from a small cohort. Consider regrouping into broader categories (e.g., luminal vs. non-luminal) to improve statistical power and interpretability.
-The observed correlation in benign lesions lacks biological context and is absent in cancer patients. Its clinical or immunological significance is questionable.
-There are inconsistencies between the p-values in the results text and Figures 2–3 (specialy in IL-22 results). Please verify and correct. Also, define the criteria for outliers/extremes used in boxplots.
-Figure descriptions repeat content already stated in the main text (e.g., for Figure 2). Could you try simplifying or omitting these sections?
- Given the limited findings, the discussion is well-cited but too detailed. A more focused and concise version would strengthen the manuscript’s impact.
-Only two cytokines are explored. Could you consider highlighting this limitation and suggesting future studies incorporating broader immune profiling (e.g., cytokine panels, PBMC analysis, receptor expression, TILs)?
I appreciate the effort and recommend revising the manuscript to focus on the exploratory and descriptive nature of the results.
Reviewer 2 Report
Comments and Suggestions for Authors
Dear
Authors
The following are a series of detailed observations aimed at improving the scientific quality, clarity, and soundness of your study.
1.- Methodology and study design:
a) The sample size is relatively small and poorly balanced across molecular subgroups. This compromises statistical power, especially for comparisons by molecular type of cancer. This should be more clearly recognized as a limitation in the discussion... so it is not valid as presented.
b) Absence of a healthy control group: The manuscript compares patients with cancer versus benign lesions but does not include a healthy control group. Given that serum cytokine levels are studied, this group would have been essential to establish a physiological baseline.
c) Although the immunological relevance of IL-21 and IL-22 is mentioned, the specific hypothesis or reason why a difference between tumor subgroups would be expected is not clear. It would be advisable to clarify this in the introduction and methodology.
2.- Results
a) The main results showed no significant differences. Although this may be due to the sample size, the manuscript emphasizes the clinical relevance of the findings without providing statistical support. It is suggested to moderate the interpretation of these results and avoid overconclusions.
b) The statistics used are basic (ANOVA/F-test), without correction for multiple comparisons or multivariate analysis. I suggest including regression models or ANCOVA if one wishes to explore associations adjusted for clinical variables.
c) Figure 2 and 3: Figures should include error bars, n per group, and a more detailed legend. Make sure the figures are self-explanatory.
3.- Discussion
a) The lack of significance is interpreted as "trends" or possible "systemic interactions," which lack quantitative support. It is important not to over-interpret harmful data.
b) The discussion mentions numerous relevant studies but mixes evidence on gene expression and serum levels, which can lead to confusion. I suggest separating molecular findings (IL-21/22 in tissue or cells) from circulating levels.
c) It would be helpful to include a comparative table with previous studies of IL-21 and IL-22 in breast cancer (comparing serum vs. tissue levels, techniques used, and the type of population) to contextualize their results better.
4.- writing and editing
a) Redundant or speculative paragraphs are identified in the discussion. Scientific editing work is recommended to improve clarity.
c) Should reflect that the findings are preliminary and without statistically significant differences.
d)d) overestimates the relevance of the results. I suggest revising the conclusion to emphasize the need for future studies with increased statistical power and well-matched comparison groups.
Other important aspects
- Authors should avoid using sentences that are too long.
- write the abstract properly
- describe the materials and methods appropriately, and use appropriate grants.
- table 1 in the materials and methods section is not appropriate. They should edit their manuscript presentation well.
- section 3 results and discussion. Authors should describe their results clearly, contrast them with existing literature, and provide conclusive findings. Authors are also advised not to be verbose.
- authors should create appropriate and orderly grants as reflected in the methodology, give context, and write appropriately.
- In good scientific writing, there are no paragraphs of less than 5 lines; this is inappropriate, as well as paragraphs that are too long.
- The images presented should be of good quality and resolution.
- it is recommended to edit, order, and justify your manuscript correctly for a good presentation.
Need to improved
Round 2
Reviewer 2 Report
Comments and Suggestions for Authors
resultado
IL-21 and IL-22 with no significant differences:
There are no statistically significant differences in serum IL-21 and IL-22 levels between groups with benign and malignant lesions, neither according to degree of malignancy nor molecular subtype. This may be due to several factors:
Possible low statistical power due to limited sample size (n=80 patients approx.).
High biological heterogeneity between patients, even within the same molecular subtype.
Serum concentrations may not accurately reflect what occurs in the tumor microenvironment (TME), where these interleukins exert their primary function.
The negative correlation between IL-21 and IL-22 in benign but not malignant lesions is intriguing.
This suggests an intact regulatory mechanism in non-cancerous tissue that is lost with the development of malignancy. It seems to be a finding with potential, although the article mentions it superficially and without further functional or clinical speculation.
discussion
1.- Excess of long and dense sentences:
2.- Lack of depth in the explanation of the negative correlation between IL-21/IL-22 in benign lesions:
This is a relevant finding and is hardly discussed. You might ask:
Is there previous evidence of this inverse relationship in non-malignant lesions?
Could it reflect an immune surveillance mechanism that is lost in cancer?
What is the potential functional or diagnostic implication of this pattern in patients?
3.- The absence of significant differences may be more a matter of statistical power than an absolute absence of biological differences. They should explicitly suggest the need for a larger cohort.
The authors do not mention whether confounding variables, such as age, body mass index, comorbidities, and previous treatments, which could affect IL-21/IL-22 levels, were considered.
5.- the manuscript does not provide conclusive evidence due to:
Limited sample size.
Lack of more detailed subgroup analysis.
Poor discussion of functional mechanisms derived from serum findings.
Still, the finding of the negative correlation in benign lesions opens up an interesting possible line of investigation: is this relationship a marker of normal immune microenvironment that breaks down in cancer? This would merit longitudinal or functional study.
Other questions :
What exact method did they use to quantify IL-21 and IL-22, ELISA, and Luminex, and what was the sensitivity/LOD?
Were confounding clinical variables such as prior therapy, age, and body mass index considered?
Was the relationship of IL-21/IL-22 with clinical parameters such as clinical stage, tumor size, or presence of metastases evaluated?
Why did you not perform a ROC or discriminatory ability analysis between benign vs. malignant using these interleukins?
Are longitudinal studies planned to assess whether these markers change during progression or treatment?
Why didn't they include a control group of healthy women to know baseline IL-21/IL-22 values?
Given the low number of cases in subgroups, did they consider an adequately powered analysis?
How do they justify the clinical significance of these markers if they did not demonstrate significant differences?
Why did they not supplement with tissue analysis (IHC or qPCR), given the significance reported in previous studies?
Why insist on a speculative discussion if your data are inconclusive?
nerd to improved
Round 3
Reviewer 2 Report
Comments and Suggestions for Authors
1.- Lack of depth in the discussion of results.
The study finds no significant differences in any of the statistical comparisons. However, the discussion does not sufficiently explore the significance of these “negative” findings.
This is important as null results can also provide insight if properly contextualized.
2.- Insufficient methodological description (in the fragment):
Although not detailed in the shared text, critical methodological information is missing: type of ELISA or method used to determine serum IL-21/22 levels, sample storage conditions, diagnostic criteria, and control for confounding variables such as age, hormonal status, or comorbidities
3.- Lack of reference to contradictory or convergent literature:
The discussion only mentions studies that point to IL-21 overexpression but does not contrast this with work indicating any correlation or with other relevant cytokines in these tumor subtypes.
4.- Rewrite the discussion with more scientific and structured language.
